# *Plasmodium chabaudi* Merozoites Obtained through a Simpler Method Do Not Survive in Classically Activated Macrophages

**DOI:** 10.3390/microorganisms12010105

**Published:** 2024-01-05

**Authors:** Pedro Souto Rodrigues, Milena de Farias Azeredo, Natália de Souza Almeida, Gisela Garcia Cabral Galaxe de Almeida, João Luiz Mendes Wanderley, Sergio Henrique Seabra, Renato Augusto DaMatta

**Affiliations:** 1Laboratório de Biologia Celular e Tecidual, Universidade Estadual do Norte Fluminense Darcy Ribeiro, Campos dos Goytacazes 28013-602, RJ, Brazil; pedrorodrigues@pq.uenf.br (P.S.R.); milenafaiff@gmail.com (M.d.F.A.); ndesouza.almeida@gmail.com (N.d.S.A.); giselagalaxe@outlook.com (G.G.C.G.d.A.); 2Laboratório de Imunoparasitologia, Universidade Federal do Rio de Janeiro, Macaé 27930-560, RJ, Brazil; lmwjoao@gmail.com

**Keywords:** *Plasmodium chabaudi*, merozoite, filtration, erythrocytes, activated macrophage

## Abstract

Malaria is caused by apicomplexan parasites of the *Plasmodium* genus. *Plasmodium chabaudi* is an excellent animal model for the study of human malaria caused by *P. falciparum*. Merozoites invade erythrocytes but are also found in other host cells including macrophages from the spleen and liver. Methodologies for obtaining merozoites usually involve treatment with protease inhibitors. However, merozoites obtained in this way may have their enzymatic profile altered and, therefore, are not ideal for cell-interaction assays. We report the obtainment of *P. chabaudi* merozoites naturally egressed from a synchronous erythrocyte population infected with schizonts forms. Merozoites had their infectivity and ultrastructure analyzed. Interaction assays were performed with mice erythrocytes and classically activated mice peritoneal macrophages, a very well-established classic model. Obtained merozoites were able to kill mice and efficiently infect erythrocytes. Interestingly, a lower merozoite:erythrocyte ratio resulted in a higher percentage of infected erythrocytes. We describe a simpler method for obtaining viable and infective merozoites. Classically activated macrophages killed merozoites, suggesting that these host cells may not serve as reservoirs for these parasites. These findings have implications for our understanding of *P. chabaudi* merozoite biology and may improve the comprehension of their host–parasite relationship.

## 1. Introduction

There were an estimated 247 million malaria cases in 2021 in 8 malaria-endemic countries [1]. There is an urgency to develop an effective vaccine or new antimalarial compounds to reduce the burden of malaria and prevent cases of drug resistance [2], and a good understanding of the molecular mechanisms involved in the infection of the human host is essential in the development of a successful treatment of malaria [3]. Since the merozoite form of *Plasmodium* spp. is extracellular and exposed to antibodies and immune cells in the bloodstream, merozoite antigens have long been regarded as attractive targets for vaccine development. There is also growing interest in the identification and development of novel drug therapies that target merozoite invasions; many of the proteins and processes required for invasion are essential and unique to the parasite. An understanding of the host–parasite relationship is essential for the development of a vaccine and for improving medications [4]. Due to the inherent obstacles to experimental work with human malaria, several animal models are used, contributing to the understanding of the disease and its treatment. Infection in mice with *P. chabaudi* is one of the closest models of disease in humans caused by *P. falciparum* [5].

One of the most relevant pathogenic processes in malaria occurs during blood infection when merozoites invade erythrocytes. Parasites develop in erythrocytes with precise periodicity, and at the end of each cycle, merozoites egress lysing erythrocytes in the process [6]. The immunopathology of severe malaria often originates from excessive immune activation by parasites. Innate immune cells of the monocyte lineage, including macrophages, play indispensable roles during malaria infection, and infected host macrophages in various organs are directly involved in the interaction with *Plasmodium falciparum*-infected erythrocytes and free merozoites, including macrophages in the liver (Kupffer cells), spleen (splenic macrophages), and bone marrow [7,8]. Indeed, the life cycle of *Plasmodium* spp. appears to be more complex, since merozoites have been found within cells other than erythrocytes and latent merozoites have been found in mice macrophages, neutrophils [9], and dendritic cells [10].

Methods based on the natural release of merozoites have been developed for malaria in humans and other primates [4,11,12]. In most of these techniques, erythrocytes infected with schizonts are concentrated and matured in vitro with the use of E64, a protease inhibitor. However, it has been shown that this compound has antimalarial effects through the binding and inhibition of falcipain-2 (FP-2), a parasite enzyme that participates in the degradation process of hemoglobin [13]. Therefore, the use of E64 may modify the enzymatic profile of the parasite. Erythrocytes with immature parasites have been separated by agglutination and merozoites were released, but low viability has been reported [12]. The use of a cell sieve permits the isolation of a large number of intact and infective merozoites of *Plasmodium knowlesi* [14]. The involved apparatus, however, is relatively complex, and this method was not considered useful for other plasmodial species, which is probably due to differences in size and deformability of the erythrocytes infected with schizonts [15]. In the case of rodent *Plasmodium* spp., free parasites can be obtained by artificial lysis of infected erythrocytes, but this kind of preparation is frequently heavily contaminated by immature forms [16]. Indeed, most of the attempts to purify merozoites that maintain their invasive capacity were unsuccessful or resulted in non-viable merozoites [4,17]. This has impaired the comprehension of many aspects of the parasite biology, including the invasion process.

In this work, we describe the obtainment of *P. chabaudi* merozoites with high purity, viability, and invasive capabilities into erythrocytes by adapting a protocol described by Chimanuka et al. 1997 [18]. These authors describe a protocol to obtain ring-infected erythrocytes that were injected in mice, resulting in animals with similar infection rates that were used to analyze total and differential parasitemia curves. However, the methodology was not used to study the biology of the obtained free merozoites, and their viability and capacity to infect other host cells were not reported. We adapted this protocol by adding a filtering step and demonstrated a highly efficient method for obtaining viable and erythrocyte-invasive *P. chabaudi*-free merozoites. Since merozoites can survive within other immune system cells [9,10], we used the merozoites obtained here to investigate whether macrophages can serve as a reservoir for the parasite and thus play a role in host infection, elucidating the findings observed in previous studies, which demonstrated the presence of merozoites within monocytes and macrophages in rodent models [9]. We also characterized the obtained merozoites and described the kinetics of the interaction with classically activated mice peritoneal macrophages to serve as a model, since merozoites normally interact with macrophages in the liver and spleen of the host.

## 2. Materials and Methods

### 2.1. Mice Infection and Natural Synchronization

Balb/c mice (6–8 weeks) were kept in a room with a 12 h light (10:00–22:00) and 12 h dark (22:00–10:00) period. *Plasmodium chabaudi* (AJ strain–lethal) was kept by inoculation of infected erythrocytes in the peritoneal cavity of mice every 6–7 days. As *P. chabaudi* is synced with the host’s melatonin, it is possible to obtain erythrocytes enriched with a specific form of the parasite depending on the hour in which the blood of the animal is collected [18]. Animal usage was approved by the Animal Ethics Committee of the Universidade Estadual do Norte Fluminense Darcy Ribeiro, protocol ID 320.

### 2.2. Obtaining Plasmodium chabaudi Merozoites

Blood was taken with a syringe with 0.1 mL of 3.7% sodium citrate by cardiac puncture 3 h before the first hour of light (7:00) for the obtainment of erythrocytes with mature schizonts, as described by Chimanuka et al. 1997 [17]. Blood taken from 2 animals (±1 mL per animal) was centrifuged at 200× *g* for 10 min, the supernatant was discarded, and 5 mL of Dulbecco’s Modified Eagle Medium (DMEM) supplemented with 10% fetal bovine serum (FBS) was added to the pellet and further incubated for 30 min at 37 °C to egress of merozoites. The solution was centrifuged at 200× *g* for 10 min, and the supernatant with free merozoites was centrifuged at 4000× *g* for 20 min in 4 microtubes (1.5 mL). The supernatant was discarded, and the pellet was resuspended in 10 mL of DMEM and then filtered (3.0 μm—Millipore filter ref. TSTP02500) or not. Because merozoites cannot be seen directly by phase of differential interference microscopy, this form was quantified by direct counting of parasites on slides obtained after cytocentrifugation. Briefly, 100 μL of the parasite suspension was mounted on slides with a cytospin and stained with Diff-Quick. Quantification was performed by counting merozoites in 5 distinct and random microscopic fields observed with the 100× objective lens. The obtained mean number of merozoites was multiplied by a factor (622) that corresponds to the number of times the area of the observed microscopic field (4.5 × 10^4^ μm^2^) fit the whole area of the cytospin (2.8 × 10^7^ μm^2^). The resulting number was the number of merozoites in 100 μL of the parasite suspension. The number of parasites was also estimated by flow cytometry through the number of events in a gate of 1 μm, which corresponds to the size of merozoites (see item 2.5).

### 2.3. Evaluation of In Vivo Viability

Balb/c mice were separated into 2 groups of 5 animals. One group received intraperitoneally 1 × 10^6^ filtered merozoites, obtained as described above. The second group was intraperitoneally injected with 1 × 10^6^ infected erythrocytes, as in the routine passage. Survival of both groups was assessed daily and was presented as a Kaplan–Meyer graph.

### 2.4. Preparation of Cytospin Slides

Glass slides were mounted with cytocentrifuge filter paper (CIENLAB) and placed in a cytocentrifuge (Fanen, mod. 218). A total of 100 μL solution containing free merozoites and infected or non-infected mouse erythrocytes were submitted to rotation at 30 g for 4 min. The slides were stained with Diff-Quick and observed by bright field microscopy.

### 2.5. Flow Cytometry Quantification and Evaluation 

Beads of 1.0, 0.5, and 0.2 μm were used to define in the side-scatter and forward-scatter dot plot a specific gate that corresponded to the estimated size of merozoites (FluoSpheres #2, Molecular Probes). Merozoites were also quantified through the corresponding number of events in this gate. Obtained merozoites, filtered or not, were incubated with propidium iodide (PI) (3 μg/mL) to assess cell viability, as PI is not membrane-permeable and binds to DNA. Positive control for PI was obtained by incubating merozoites in DMEM containing 0.5% Triton X-100 for 15 min. After 5 min, cells were assessed on a FACScalibur flow cytometer. Data were analyzed in the FlowJo X10.0.7 program.

### 2.6. Erythrocyte Infection

Blood from Balb/c mice was obtained by cardiac puncture into syringes with 0.1 mL of 3.7% sodium citrate. Blood was centrifuged and erythrocytes were washed with 10 mL of RPMI-1640 supplemented with 20 mL 1M HEPES buffer, 5 mg hypoxanthine, 2 mg gentamicin, 10 mL of 200 mM L-glutamine, and 42 mL of 5% sodium bicarbonate solution per liter. The erythrocyte suspension was supplemented with 10% mouse serum obtained from adult Swiss mice. Then, 1 × 10^7^ and 0.5 × 10^7^ merozoites were added to 1 × 10^7^ erythrocytes in a final volume of 2 mL per tube. Samples (100 μL) were obtained after 30, 60, and 120 min and cytospins were prepared. Erythrocytes with and without ring forms were counted in duplicate cytospins in three independent experiments to determine the infection index. A total of 300 cells per cytospin were counted.

### 2.7. Activation of Peritoneal Macrophages and Interaction with Parasites

Murine peritoneal macrophages were obtained as described by Seabra et al. (2004) [19]. Briefly, macrophages were obtained by peritoneal lavage (5 mL of acidified DMEM, pH = 5) of Balb/c mice that had their peritoneal cavity stimulated 4 days earlier with 1 mL of 3% sodium thioglycolate. Macrophages were centrifuged, counted, and plated on glass coverslips on 24-well culture plates. After 1 h of adhesion at 37 °C, cells were washed with phosphate-buffered saline (PBS) at 37 °C and cultivated for 24 h with DMEM containing 5% FBS, lipopolysaccharide (LPS) (100 ng/mL), and interferon-γ (IFN-γ) (50 U/mL) at 37 °C in a 5% CO_2_ atmosphere. After 24 h, cells were washed and merozoites were added to macrophages in a 10:1 ratio between parasites and macrophages. After 2 h, cells were washed with PBS; some cells adhered to coverslips and were collected, and the remaining cells were further cultivated for 24 and 48 h in the same culture medium.

### 2.8. Optical Microscopy and Quantification of the Infection

Cells adhered to coverslips were fixed in a 4% formaldehyde PBS solution, stained with Giemsa, dehydrated in acetone-xylol solutions, and mounted on slides with Entellan. Cells were observed by bright field and polarized microscopy in a Zeiss Axioplan microscope. Parasite development was evaluated by counting infected and non-infected macrophages and merozoites numbers per infected cell. At least 300 macrophages per coverslip were counted. Each time point was evaluated in triplicate in three independent experiments.

### 2.9. Production of Anti-Merozoite Polyclonal

An anti-merozoite polyclonal antibody was obtained using a standard immunization protocol. Mice (weighing about 20 g) that were previously infected for 4 months with *P. chabaudi* (AS strain–non-lethal) CB57BL/6 were inoculated intraperitoneally with 200 μL of PBS containing 1 × 10^5^ dead merozoites. Merozoites were obtained as described above, filtered, quantified, and submitted to three cycles of freeze and thaw. Animals were inoculated 4 times, once a week. In the first 2 inoculations, 1 mg of Al(OH)_3_ per animal was used as an adjuvant. In the 5th week, animals were euthanized, their blood removed, and their plasma obtained. The presence of anti-merozoites antibodies was assayed by immunolocalizing merozoites within macrophages.

### 2.10. Immunofluorescence

After interactions for 2 and 24 h of *P. chabaudi* merozoites with activated macrophages, the cells were fixed with 4% formaldehyde in PBS, pH 7.4. Cells were permeabilized with 0.5% Triton X-100 in PBS and incubated with ammonium chloride (100 mM in PBS) and 3% bovine serum albumin (BSA–Sigma, St. Louis, MO, USA) in PBS buffer (BSA-PBS). Cells were incubated with an anti-merozoite antibody diluted 1:800 in BSA-PBS for 1 h. Cells were washed with PBS, incubated for 10 min with BSA-PBS, and incubated with TRITC anti-mouse (Sigma, St. Louis, MO, USA) diluted 1:400 in BSA-PBS for 1 h. Cells were washed and mounted with ProlongGold with DAPI (Thermo Fisher Scientific, Waltham, MA, USA) and observed in a Zeiss fluorescence Axioplan microscope equipped with epifluorescence illumination and an HBO100 mercury lamp. Images were captured and processed in Adobe Photoshop CS3, VN 10.

### 2.11. Preparation of Samples for Transmission Electron Microscopy

Merozoite suspension was fixed with 2.5% glutaraldehyde and 4% recently prepared formaldehyde in sodium cacodylate buffer (0.1 M, pH 7.2) for 2 h. Merozoite-infected macrophages cultured in 25 cm^2^ culture flasks were also fixed as above and removed from the flasks via cell scraping. Cells were washed with sodium cacodylate buffer and post-fixed with 1% osmium tetroxide, 1.6% potassium ferrocyanide, and 5 mM calcium chloride in sodium cacodylate buffer for 1 h. Cells were dehydrated with acetone serial concentrations of 30%, 40%, 50%, 70%, and 100%. Inclusion was performed with epoxy resin. Ultrafine sections were contrasted with uranyl acetate and lead citrate and observed in a Jeol JEM-1400 Plus transmission electron microscope.

### 2.12. Statistical Analysis

Experiments were performed at least three times, as indicated in the figure legends. The results were expressed as means and standard deviation. The variances between three independent experiments were analyzed using the one-way analysis of variance (ANOVA), followed by Tukey’s multiple comparisons post-test in the Graph-Pad Prism 8.0 statistical program. *p* < 0.05 values were considered significant.

## 3. Results

In the protocol from Chimanuka et al. 1997 [18], the obtained merozoites were mixed with a great number of erythrocytes and leukocytes (Figure 1a), but the additional filtering step was able to remove all leukocytes and most erythrocytes (Figure 1b). This was further confirmed by flow cytometry analysis where the number of events corresponding to merozoites increased with filtration and the number of events with higher FSC profiles was mostly absent (Figure 1e,f). We used beads of known sizes to define a gate that corresponded to the size of merozoites. This confirmed that the events that appeared in the set gate were mostly merozoites. Merozoites presented high viability after incubation with PI (Figure 1g), which was not altered after the filtration step (Figure 1h). Viability of merozoites was confirmed through negative staining with PI (Figure 1c). Merozoites with their membrane lysed using Triton X-100 served as a positive control for PI (Figure 1d). Furthermore, the infectivity of the merozoites was also confirmed, as the obtained forms were able to infect and kill mice (Figure 2).

Transmission electron microscopy analysis showed a homogeneous population of merozoites after filtration with the presence of some clusters of hemozoin (Figure 3a). Merozoite presented well-preserved organelles. The apical complex was clearly seen, as were the inner membrane complex, rhoptry, dense granules, mitochondrion, nucleus, and ribosomes (Figure 3b).

Merozoites obtained after filtration were able to infect erythrocytes and were established as ring forms (Figure 4a). Filtered merozoites were able to infect erythrocytes after 30 min of interaction (Figure 4b,c). Quantification of the invasion by post-invasion parasitemia demonstrated that longer periods of infection did not increase the infection rate of erythrocytes (Figure 4b,c). The efficiency of merozoite invasion was inversely proportional to the merozoite:erythrocyte ratio as the lower ratio led to a higher percentage of infected erythrocyte (Figure 4b,c). 

Merozoites were found in activated macrophages 2 h post-infection (Figure 5a), and hemozoin was seen by polarized microscopy (Figure 5b). During the course of the infection, the amount of merozoites decreased in the macrophages (Figure 5c,e). After 48 h post-infection, it was difficult to find merozoites in macrophages, but hemozoin was visible (Figure 5e,f). Quantification of the infection revealed a drastic decrease in infected macrophages (Figure 5g) and in the mean number of merozoites per macrophage (Figure 5h).

Immunofluorescence revealed that mice chronically infected with *P. chabaudi* of the AS strain produced a polyclonal antibody that labeled merozoites in macrophages. However, naïve animals that received the same inoculations as chronically infected mice did not produce antibodies capable of labeling merozoites. Uninfected macrophages were negative for merozoite labeling (control), showing that the antibody produced was specific. Interestingly enough, the polyclonal antibody 2 h post-infection labeled merozoites inside macrophages with a clear round pattern (Figure 6a–c), but labeling after 24 h of infection changed to a fuzzy pattern (Figure 6d–f). Analysis by transmission electron microscopy of infected macrophages for 24 h shows several vacuoles in the cytoplasm and no merozoites (Figure 7a,b), indicating that merozoites were degraded at this period of infection. Membrane profiles were seen inside some of these vacuoles (Figure 7b).

## 4. Discussion

Malaria is one of the most prevalent parasitic diseases in tropical countries [1]. Several studies describe different procedures to obtain *Plasmodium* spp. merozoites. It has been reported that merozoites obtained by schizont rupture and differential centrifugation have low viability. It has been pointed out that the poor viability after schizont rupture may be the main reason why attempts to isolate viable and infective merozoites are often unsuccessful [4]. Merozoites obtained by these methods are usually used long after schizont rupture and involve several handling and washing steps, in addition to the use of enzymatic inhibitors [4,17]. Furthermore, merozoites are not restricted to erythrocytes and have been described in other host cells such as dendritic cells [10], neutrophils, and macrophages [9], indicating that the interaction with these cells is relevant. Here, we describe a chemical-free and simple methodology to obtain infective merozoites to study distinct aspects of their biology. We reported the obtainment of highly purified merozoites with excellent viability and preserved ultrastructure. Filtered merozoites were capable of infecting erythrocytes. In addition, it was demonstrated that the infective merozoites could not survive in an activated macrophage model, suggesting that activated macrophages from key organs may control the survival of these parasites.

A simple and widely used method for obtaining free viable and infectious merozoites has been described [4]. However, their protocol uses E64, a protease inhibitor, which may affect the enzymatic profile of the parasite as it inhibits the activity of FP-2 [13] an enzyme of the cysteine proteases group that degrades host hemoglobin, which is vital for the parasite survival in the host [20]. Therefore, the use of E64 may alter the activity of other cysteine enzymes of the parasite. For these reasons, merozoite obtained with methodologies using E64 should be avoided for invasion tests and chemotherapy studies. Here, the obtained merozoites were physically filtered, but the viability of the parasites was preserved. We chose not to use a 1.2 µm filter as in other studies, as it is described that the filtering process often greatly decreases the viability of merozoites [9]. In fact, using the 3 µm filter, we were able to isolate merozoites that preserve their viability. Furthermore, merozoites in suspension were pelleted by centrifugation resulting in agglutination of these cells. Agglutination of merozoites by centrifugation has been reported before [4] but did not alter the infectivity of these forms. This is in agreement with the merozoites obtained here, which were capable of infecting erythrocytes, indicating that their infectivity was not altered by this agglutination. Transmission electron microscopy of the filtered merozoites showed a homogeneous population with a well-preserved ultrastructure. As expected, it was possible to see hemozoin clusters in the merozoite preparations. During schizont formation, hemozoin is concentrated in a food vacuole in the center of this form [21], and when rupture occurs, this food vacuole containing hemozoin is released. Further purification steps, such as magnetic column usage [22,23], could be used to remove hemozoin from this preparation.

Merozoites were able to infect erythrocytes, as ring forms were clearly seen. Higher erythrocyte post-invasion parasitemia was obtained with the lower merozoite:erythrocyte ratio. O’Donovan and Dalton (1993) [24] obtained similar results using a different strain of *P. chabaudi*. However, they performed an assay using naturally infected erythrocytes in co-culture with uninfected erythrocytes. Thus, it was not possible to determine the exact number of merozoites that invaded the added uninfected erythrocytes. Boyle et al. 2010 [4] also obtained a similar result with merozoites of *P. falciparum,* whereby increasing the merozoite:erythrocyte ratio results in a lower erythrocyte infection rate. They suggest that a period of competitive exclusion could limit the invasion of erythrocytes, supporting that efficient invasion may be limited by an excess of merozoites. Such a hypothesis also seems to apply to *P. chabaudi* merozoites, as demonstrated here. The erythrocyte invasion assay described here may serve as an interesting methodology to study the different aspects of the merozoite capacity to infect these cells, allowing testing of invasive blocking compounds and a better understanding of the invasive biology of these forms.

The current view is that, in mammals, the conversion to merozoites occurs only in hepatocytes and red blood cells; however, some studies already document the transformation of sporozoites of rodent-infecting *Plasmodium* into merozoites in the skin of mice [25]. A previous study investigating blood-stage parasites of rodent *Plasmodium* spp. report the existence of sacks of merozoites within macrophages and neutrophils [9]. Interestingly, during *P. yoelii* mice infection, macrophages of the spleen contained vesicles with up to 40 merozoites [26]. Thus, cells other than erythrocytes were shown to serve as hosts to blood-stage parasites, and possibly they were in a latent stage in these cells. This demonstrates that the life cycle of *Plasmodium* spp. is more complex than we previously thought. Therefore, we decided to investigate whether macrophages could serve as a reservoir for the parasite, similar to dendritic cells [10]. If this hypothesis is confirmed, it would explain the results of previous studies that found merozoites inside macrophages in rodents [9]. This hypothesis is reinforced by studies that indicate that some intracellular parasites, including *Toxoplasma gondii*, can alter the immune response of macrophages by down-regulating the enzyme inducible nitric oxide synthase (iNOS) [19]. To assess whether this modulation mechanism is responsible for finding merozoites inside macrophages in some studies, we infected activated mice peritoneal macrophages, a classic and well-established model that expresses iNOS, to test if these cells may serve as a reservoir for merozoites of *P. chabaudi*. After long periods, post-infection merozoites were not seen in these activated macrophages, and only hemozoin was detected by polarized microscopy. In addition, the generated polyclonal antibody clearly labeled merozoites in macrophages after 2 h post-infection, but merozoite labeling changed to a fuzzy pattern after long post-infection periods, suggesting parasite degradation followed by dispersion of their antigens. This degradation was further confirmed by transmission electron microscopy, where no merozoites were observed in activated macrophages after 24 h post-infection. Therefore, we concluded that merozoites were not able to persist in activated mice peritoneal macrophages. This may be due to high activation by IFN-γ and LPS of macrophages obtained from a mouse peritoneal cavity stimulated with thioglycolate, resulting in macrophages that were too microbicidal. Another possible explanation for the high microbicidal activity of macrophages is the presence of hemozoin. Hemozoin is released into the circulation following erythrocyte lysis and merozoite replication. Once in the bloodstream, hemozoin is phagocytosed by macrophages [27]. Merozoites obtained through our methodology are not hemozoin-free, and it has been reported that hemozoin may have a protective role in the infection, through modulation of the microbicidal activity of macrophages [28]. Alternatively, to explain results where merozoites are found latently within phagocytes, several studies have shown that *Plasmodium* spp. is intimately linked with the host spleen [29,30,31] and merozoites may be adapted to deal with macrophages from this organ. In addition, the parasite of the apicomplexa phylum *Toxoplasma gondii* has a protein known as an inhibitor of STAT1-dependent transcription, which is responsible for blocking IFN-γ dependent transcription, avoiding activation of host cells [32,33]. Further studies are needed to confirm whether *P. chabaudi* has a similar mechanism that prevents host macrophage activation after being infected, explaining the results obtained by Landau et al. 1999 [9].

## 5. Conclusions

In conclusion, our study represents significant advancements in understanding the biology of *Plasmodium chabaudi* merozoites, a valuable animal model for investigating human malaria caused by *P. falciparum*. By obtaining merozoites through a natural egress from synchronized populations of erythrocytes infected with schizont forms, we have developed a streamlined and effective approach compared to conventional methods employing protease inhibitors. We demonstrated that these merozoites retain their infectivity and ultrastructure, effectively infecting erythrocytes and inducing mortality in mice.

Surprisingly, we observed that a lower merozoite-to-erythrocyte ratio resulted in a higher percentage of infected erythrocytes, highlighting crucial nuances in the dynamics of this parasite–host interaction. Furthermore, our interaction assays revealed that merozoites were susceptible to the action of classically activated peritoneal macrophages, suggesting that these host cells may not serve as reservoirs for these parasites.

These findings not only provide valuable insights into the biology of *P. chabaudi* merozoites but also have significant implications for the broader understanding of the host–parasite relationship. By offering a simpler and reliable approach for obtaining viable merozoites, our method may facilitate future research and contribute to the development of effective strategies in the fight against malaria. Future research should delve deeper into the broader host–parasite relationship, considering additional cell types and the intricate immune responses involved. 

## Figures and Tables

**Figure 1 microorganisms-12-00105-f001:**
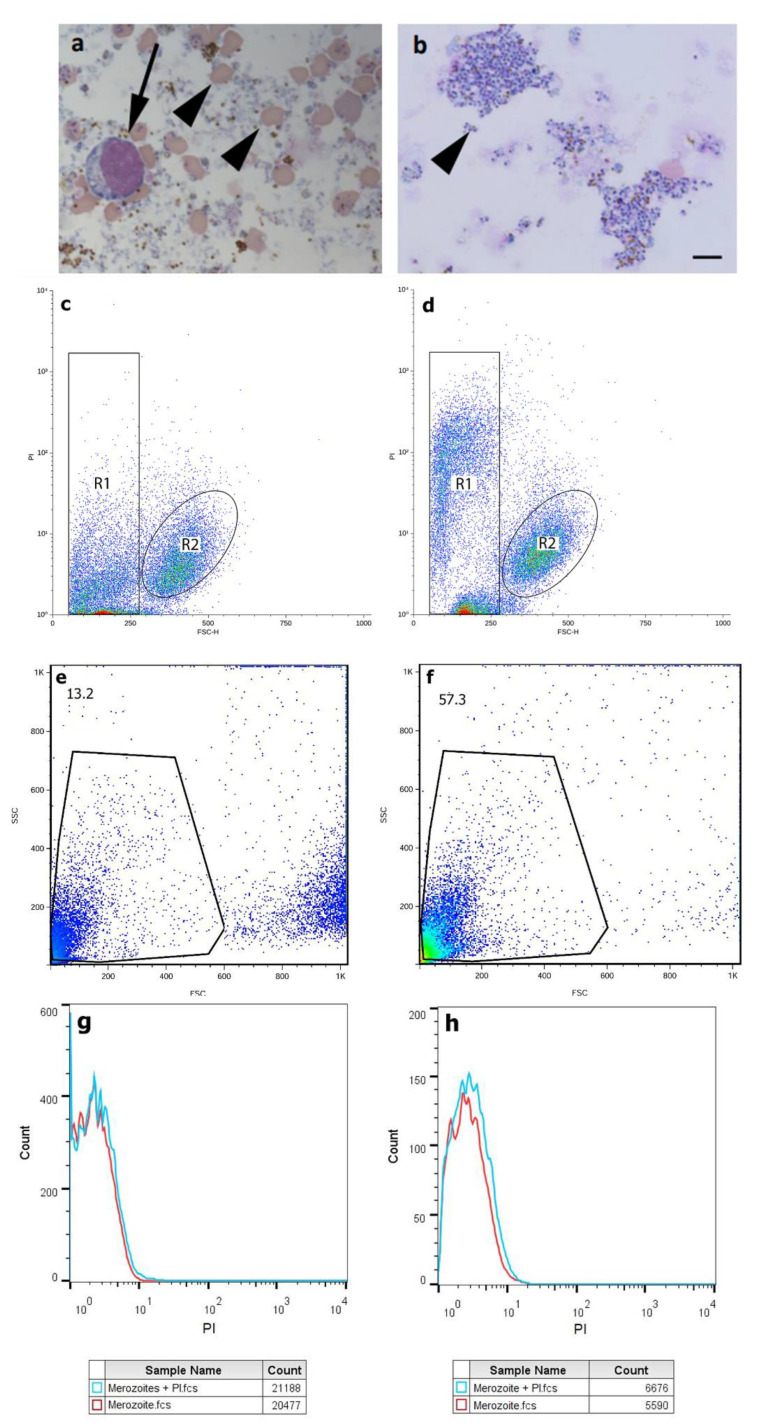
Bright field microscopy of cytospin slides and flow cytometry of obtained *Plasmodium chabaudi* merozoites. (**a**) Cytospins slides of obtained merozoites without filtration presented leukocytes (arrow) and erythrocytes (arrowheads). (**b**) Cytospins of filtered merozoites presented enriched parasites (arrowhead). Bar = 5 μm for “a” and “b”. Flow cytometry PI and forward-scatter dot plots of freshly obtained merozoites (**c**) and merozoites incubated with Triton X-100 for 15 min (**d**); Triton X-100 causes a significant increase in the PI signal at the merozoite gate (R1), without an expected change at the erythrocytes gate (R2). Flow cytometry side-scatter and forward-scatter dot plots of unfiltered (**e**) and filtered (**f**) merozoites. Percentages of events are depicted above the gate. Cell viability analysis by flow cytometry after PI labeling of unfiltered (**g**) and filtered (**h**) merozoites. Experiments were repeated three times.

**Figure 2 microorganisms-12-00105-f002:**
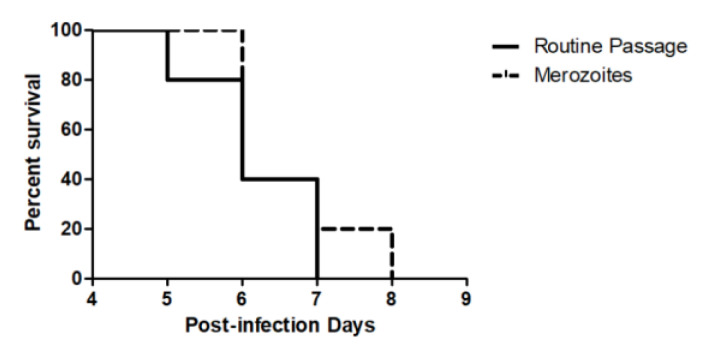
Survival of mice infected with Plasmodium chabaudi. Mice were infected with filtered merozoites or with infected erythrocytes obtained from infected mice routinely used to maintain the parasite. No difference was detected between both groups (*n* = 5 per group). The experiment was performed three times.

**Figure 3 microorganisms-12-00105-f003:**
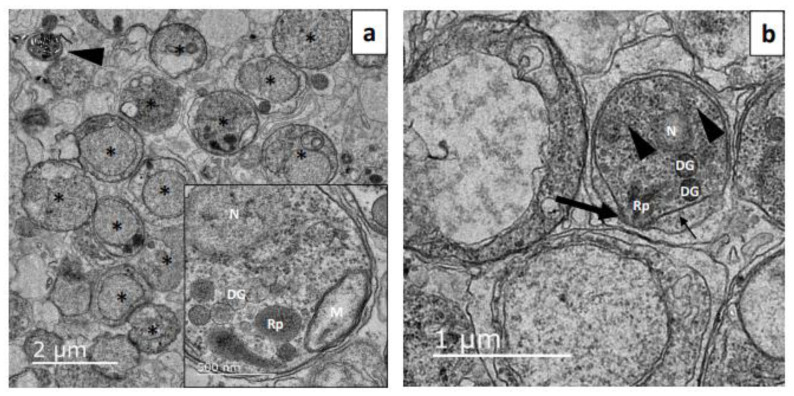
Transmission electronic microscopy images of filtered merozoites of *Plasmodium chabaudi*. (**a**) Obtained merozoites (asterisk) presented preserved ultrastructure, and a hemozoin cluster was seen (arrowhead). (**b**) The apical complex (large arrow) can be seen as the inner membrane complex (arrow) and ribosomes (arrowheads). N—Nucleus, Rp—Rhoptry, DG—Dense Granules, M—Mitochondrion. The experiment was performed three times.

**Figure 4 microorganisms-12-00105-f004:**
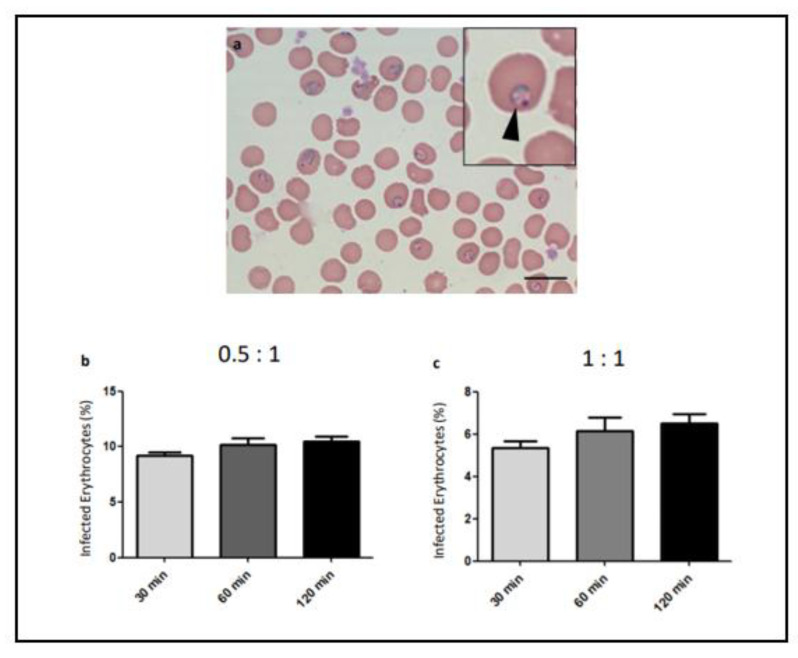
Erythrocyte invasion assay with filtered merozoites of *Plasmodium chabaudi*. Erythrocyte suspension was incubated with filtered merozoites for 30, 60, and 120 min, cytospins slides were prepared. (**a**) Merozoites were able to invade erythrocytes and develop into ring forms (arrowhead) after 120 min. Bar = 15 μm. Quantification of the percentage of infected erythrocytes after invasion using 0.5:1 (**b**) or 1:1 (**c**) merozoites:erythrocyte ratios. The experiment was performed three times.

**Figure 5 microorganisms-12-00105-f005:**
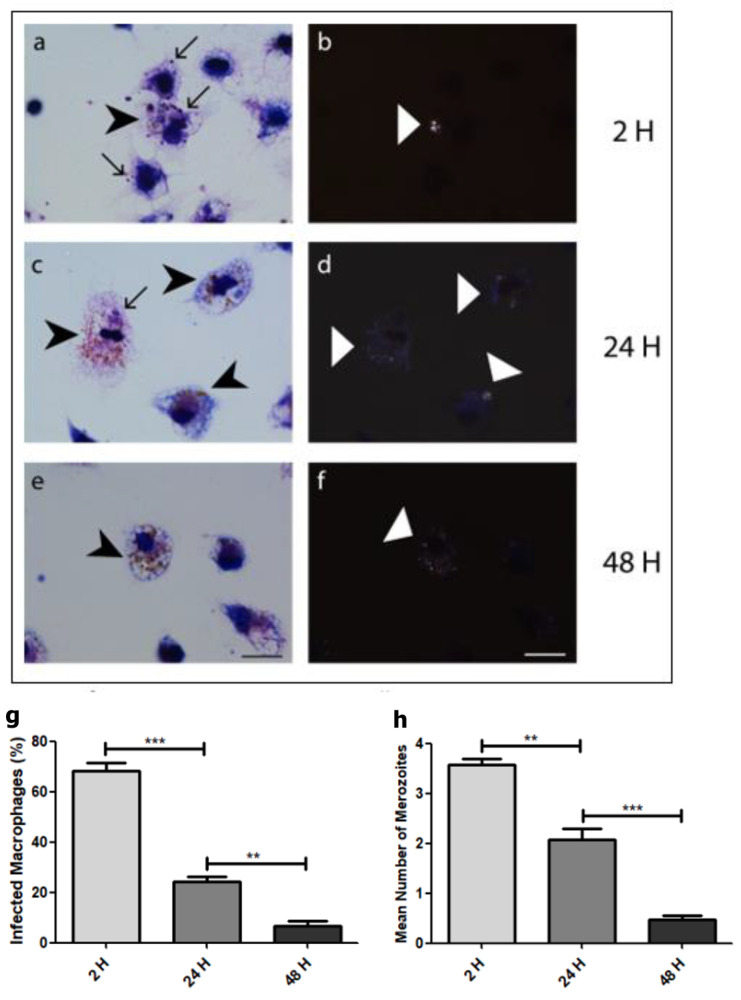
Bright field microscopy of Giemsa stained (**a**,**c**,**e**) and polarized light microscopy (**b**,**d**,**f**) of activated mice peritoneal macrophages infected with *Plasmodium chabaudi* filtered merozoites for 2 h (**a**,**b**), 24 h (**c**,**d**), and 48 h (**e**,**f**). After 2 h of infection, merozoites (arrows) and hemozoin (black and white arrowhead) are seen. Only hemozoin can be seen after 48 h (white arrowhead). Bar = 15 μm for all figures. Quantification of the percentage of infected macrophages (**g**) and the mean number of merozoites per macrophage (**h**). Significantly different by one-way analysis of variance: ** *p* < 0.01, *** *p* < 0.001. The experiment was performed three times.

**Figure 6 microorganisms-12-00105-f006:**
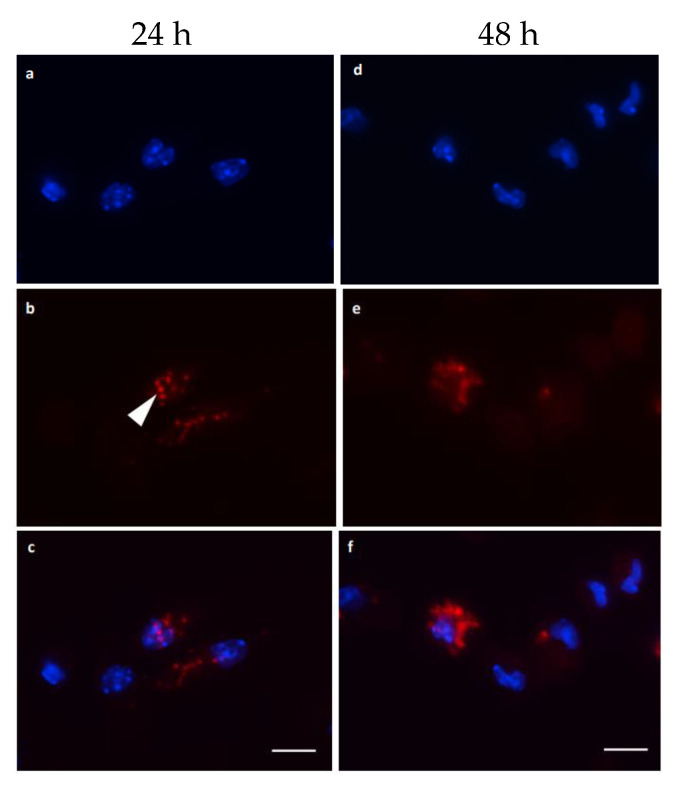
Immunolocalization of *Plasmodium chabaudi* filtered merozoites after infection of activated mice peritoneal macrophages. The produced polyclonal antibody was used to label merozoites in macrophages after 2 h (**a**–**c**) and 24 h (**d**–**f**) of infection. (**a**,**d**) DAPI (blue); (**b**,**e**) antibody labeling (red); (**c**,**f**) merge. Labeling changed from a round (arrowhead) to a fuzzy pattern after 24 h of infection. Bar = 20 μm. The experiment was performed three times.

**Figure 7 microorganisms-12-00105-f007:**
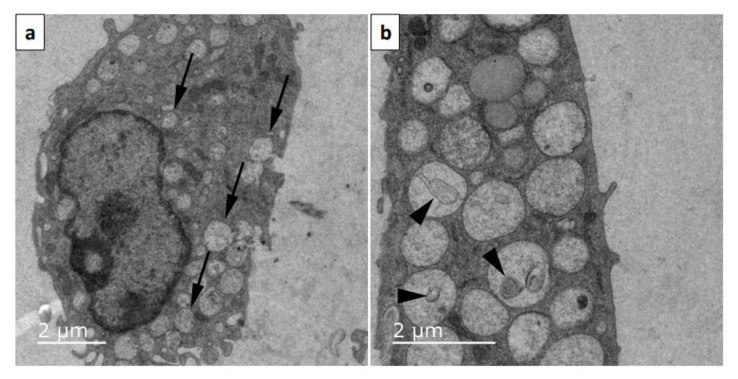
Transmission electronic microscopy of activated macrophages infected for 24 h with *Plasmodium chabaudi* filtered merozoites. (**a**) Vacuoles can be seen throughout the cell cytoplasm (arrows). (**b**) Some of these vacuoles contained membrane profiles (arrowheads). The experiment was performed three times.

## Data Availability

Data are contained within the article.

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
