# Peer review of "Plasmodium chabaudi* Merozoites Obtained through a Simpler Method Do Not Survive in Classically Activated Macrophages"

_microorganisms, 2024, doi:10.3390/microorganisms12010105_

Round 1
Reviewer 1 Report
Comments and Suggestions for Authors
The manuscript entitled "Plasmodium chabaudi merozoites obtained through a simpler method do not survive in classically activated macrophages" Title, abstract and overall rationale of work is written satisfactory. Still, there are major concerns, which needs to be addressed before publication.
1) Abstract is written well and describe concise way.
2) Introduction section: This section is written lengthy and I suggest author to reduce specially line no. 48-52.
3) Material method section: 2.5 section author need to write details about the flow cytometry to understand clear methodology.
4) In the result section: figure 1 c, d where is control group. How many times author repeat this experiments please write in the below.
5) Figure 5 author need to increase resolution of this figure.
6) For results section author need to describe more details because I identified author did many experiment but they describe very less.
7) Discussion section: This section author need to improve because author written the results part but they do not discuss properly and I saw the lack of discussion in this manuscript. I recommend author, they should elaborate the discussion part and author need to revise and compare the study with relevant study.
8) Conclusion: This section written almost same as abstract section. Author must be write proper conclusion section, limitation of study and future prospective.
9) Some references are too long and author need to revise. I suggest author to revise if other latest manuscript is available in the same information.
Comments on the Quality of English LanguageQuality of English is good.
Reviewer 2 Report
Comments and Suggestions for Authors
This is a manuscript reporting on the isolation and characterization of Plasmodium chabaudi merozoites. This is an important technique; the ability to infect erythrocytes synchronously is a valuable tool and viable isolated merozoites are key to that goal. There have been several protocols described in the past for the isolation of viable merozoites in several of the Plasmodium species. As the authors correctly point out – many of these take advantage of protease inhibitors that prevent the rupture of late stage schizonts – allowing for collection of this life stage with subsequent mechanical or enzymatic treatment of these schizonts which are ‘poised’ for merozoite release. A problem inherent to these techniques is the possible cross-reactivity of the inhibitors used – perhaps leading to inhibition of merozoite function. The authors take advantage of the light synchronicity of P. chabaudi in Balb/C mice to devise a new technique. The synchronous nature of the infection was utilized to harvest parasites just as they are ready to burst and release merozoites. Coupled with a filtration step, the authors can produce a relatively pure population of merozoites. They go on to show that these merozoites appear normal by IHC and EM. Furthermore, they can invade erythrocytes and kill mice upon direct injection. The viability and functionality of the merozoites is well documented. They couple these findings with the findings that these parasites are not viable in activated macrophages. This coupling of a new method to produce merozoites and their susceptibility to endocytosis by activated macrophages is not clear. As presented, these appear to be two important yet unrelated scientific findings. My major critique of the manuscript would be a need for the integration of these two findings. The authors discuss some as to the possibility of the macrophage serving as a reservoir for merozoites, but this is not well discussed. How would this work in the life cycle. Would merozoites develop in the Macrophage? How long would one expect a merozoite to survive in a macrophage and what would be the result? Would it then be released to invade an erythrocyte? More discussion is needed as to why the interactions between merozoites and macrophages is important in the pathophysiology of the disease.
Other concerns include:
The lack of a positive control in Figure 1. Firstly, the use of PI as a measure of parasite death needs to be explained, and then a culture or sample with significant death needs to be used to show that the PI staining and detection is functional in the system.
Figure 4 has remarkably tight error bars. No information on the number of technical or biological repeats is given. These details should be included.
Page 368-370 – explanation of Mz:Erythro ratio is still not very clear. The authors need to be careful in defining “invasion rate”. In the referenced paper (Boyle et al) invasion rate is defined as “the proportion of merozoites that invaded”. That is not how the authors of this paper are defining invasion rate. The invasion rate here is comparable to the “post-invasion parasitemia” that is used in Boyle et al.
Smaller concerns:
Line 47 – Are hundreds of merozoites truly released in P. chabaudi? In P. falciparum, 32 seems to be the maximum released per schizont.
Line 93 – Balb not balc
Line 302 – I suspect that the authors mean “fuzzy” not “fussy”
Comments on the Quality of English LanguageSmall edits in word choice and grammar are needed.
Round 2
Reviewer 1 Report
Comments and Suggestions for Authors
The authors have addressed all the concerns raised in the previous version of the manuscript and the quality has much improved after incorporating required modifications. Therefore, the manuscript may be considered for publication in this Journal.
Author Response
We thank the reviewer for revising our manuscript.
Reviewer 2 Report
Comments and Suggestions for Authors
The report has been substantially improved and all of my concerns have been addressed. A very minor concern is that the added sentence on line66-67 does not read well. I believe that the phrase "This way" should be removed.
Author Response
We thank the reviewer for revising our manuscript. The sentence on line 66-67 was deleted.